# Astaxanthin-Loaded Pickering Emulsions Stabilized by Nanofibrillated Cellulose: Impact on Emulsion Characteristics, Digestion Behavior, and Bioaccessibility

**DOI:** 10.3390/polym15040901

**Published:** 2023-02-11

**Authors:** Supaporn Saechio, Ploypailin Akanitkul, Parunya Thiyajai, Surangna Jain, Nattapol Tangsuphoom, Manop Suphantharika, Thunnalin Winuprasith

**Affiliations:** 1Institute of Nutrition, Mahidol University, Nakhon Pathom 73070, Thailand; 2Department of Food Science and Technology, University of Tennessee, Knoxville, TN 37996, USA; 3Department of Biotechnology, Faculty of Science, Mahidol University, Rama 6 Road, Bangkok 10400, Thailand

**Keywords:** Pickering emulsion, astaxanthin, nanofibrillated cellulose, in vitro digestion

## Abstract

Astaxanthin (AX) is one of the major bioactives that has been found to have strong antioxidant properties. However, AX tends to degrade due to its highly unsaturated structure. To overcome this problem, a Pickering O/W emulsion using nanofibrillated cellulose (NFC) as an emulsifier was investigated. NFC was used because it is renewable, biodegradable, and nontoxic. The 10 wt% O/W emulsions with 0.05 wt% AX were prepared with different concentrations of NFC (0.3–0.7 wt%). After 30 days of storage, droplet size, ζ-potential values, viscosity, encapsulation efficiency (EE), and color were determined. The results show that more stable emulsions are formed with increasing NFC concentrations, which can be attributed to the formulation of the NFC network in the aqueous phase. Notably, the stability of the 0.7 wt% NFC-stabilized emulsion was high, indicating that NFC can improve the emulsion’s stability. Moreover, it was found that fat digestibility and AX bioaccessibility decreased with increasing NFC concentrations, which was due to the limitation of lipase accessibility. In contrast, the stability of AX increased with increasing NFC concentrations, which was due to the formation of an NFC layer that acted as a barrier and prevented the degradation of AX during in vitro digestion. Therefore, high concentrations of NFC are useful for functional foods delivering satiety instead of oil-soluble bioactives.

## 1. Introduction

Astaxanthin (AX), one of the bioactive compounds of the carotenoid family, contains both a hydroxyl (–OH) and carbonyl (C=O) group on either end of the molecule, and are naturally found in the *trans*-configuration. Typically, AX can be observed in microalgae, salmon, trout, krill, shrimp, crabs, and crustaceans [1,2], particularly in the green microalga *Haematococcus pluvialis* [3]. Owing to its antioxidant properties, AX exhibits therapeutic effects on age-related diseases, such as, Alzheimer’s, Parkinson’s, and Huntington’s, which are mainly caused by neuroinflammation and oxidative stress. For this reason, AX is used in multi-target drug delivery for preventing the progression of these diseases [4,5]. However, due to its highly unsaturated structure, AX is susceptible to degradation when exposed to oxygen, light, and high temperatures during processing, transportation, and storage. The poor chemical stability and water solubility of AX limit its applications in the food industry [6]. Therefore, numerous attempts have been made to improve the stability, solubility, and bioaccessibility of AX using encapsulation technology. Emulsion-based delivery systems are one of the most suitable methods for the encapsulation of lipophilic bioactive compounds, such as AX. Bioactive-loaded emulsions systems can be modestly prepared by dispersing oil-soluble bioactive compounds in an oil phase and homogenizing this with a continuous aqueous phase containing an emulsifier to stabilize the system [7,8]. Moreover, oil-in-water (O/W) emulsions are widely used for encapsulating oil-soluble bioactive compounds to improve their water dispersibility, leading to better utilization for food applications. However, O/W emulsions have a limited ability to control the chemical stability of AX because the emulsifier only forms a thin layer around the oil droplets to stabilize the emulsion. This stability of O/W emulsions tends to possess a short shelf-life and leads to flocculation during the storage of the emulsions [9]. Previous studies [10,11,12,13] have shown that a number of methods for encapsulating oil-soluble vitamins and bioactives, including vitamins E and D, curcumin, and β-carotene, have been investigated to prolong their stability. Therefore, the development of delivery systems to improve the encapsulation efficiency of oil-soluble bioactives along with suitable emulsifiers needs to be investigated to improve the protection and delivery of bioactive components in foods [14,15].

Another challenge is to improve the emulsifier, which mainly consists of monoglycerides, proteins, phospholipids, and polysaccharides that are able to adhere to the oil–water interface and prevent the emulsion droplets from aggregating [16,17]. In this study, Pickering or particle-stabilized emulsions were prepared because they exhibit high stability to droplet aggregation compared to conventional molecular emulsifiers [18,19]. However, the challenge in the preparation of Pickering emulsions is to obtain suitable solid particles that adsorb at the oil–water interfaces [20]. Nanofibrillated cellulose (NFC), a plant-derived nanocellulose, is extracted from various plant sources, including wood, cotton, flax, or hemp [21,22]. At present, NFC can be used in different sectors in the food industry, for instance, stabilizing agents, emulsifiers, functional ingredients, or a carrier agents for bioactive compounds [23,24]. NFC is a potential emulsifier because it has an amphiphilic surface nature that originates from the hydrophobic face and hydrophilic edge of cellulose chains [23], and a long fibrous structure that potentially forms a 3D network after dispersing in the water phase of emulsions [25]. Therefore, the emulsions stabilized with NFC tend to have a high yield stress that is able to resist the instability of emulsions, e.g., coalescence, creaming, or sedimentation. Therefore, NFC as an emulsifier may play an important role in improving the Pickering emulsion’s stability [26,27,28].

Previous studies [28,29] have investigated the effects of the encapsulation of oil-soluble bioactives on their bioaccessibility and storage stability using NFC as an emulsifier. Burgos-Díaz et al., 2020, also prepared Pickering emulsions using a protein isolate from the protein-rich lupin variety (Lupinus luteus). However, there is no study on the encapsulation of AX in Pickering emulsions using NFC as an emulsifier with the potential to provide a protective environment to prevent the degradation of AX in the food and gastrointestinal tracts. Therefore, the aim of this study is to formulate emulsion-based delivery systems of AX to improve their stability and bioaccessibility during an in vitro study. The whey protein isolate (WPI)-stabilized emulsion is used as a control in this study. Thus, in this study, the physical and chemical properties and gastrointestinal fate of the NFC-stabilized emulsion are investigated to improve the stability and bioaccessibility of AX, which can be used for the development of food products with great health benefits.

## 2. Materials and Methods

### 2.1. Materials

Astaxanthin (AX) (>97%), mucin from porcine stomach (Cat# M2378), pepsin from porcine gastric mucosa type II (Cat# P6887), porcine lipase (Cat# SRE0049), porcine bile extract (Cat# B8631), and Nile red were purchased from Sigma-Aldrich (Sigma-Aldrich, Inc., St. Louis, MO, USA). Sodium azide was purchased from Ajax Finechem (Thermo Fisher Scientific Ltd., Auckland, New Zealand). A suspension of 3 wt% nanofibrillated cellulose (NFC) was purchased from Cellulose Lab (Cellulose Lab Inc., Fredericton, NB, Canada). The TEM image of NFC used in this study was investigated and shown in our previous research [30]. Soybean oil was purchased from a local supermarket and used without further purification. Whey protein isolate (WPI), under the tradename Provon 292, was purchased from Glanbia Nutritionals, Inc. (Carlsbad, CA, USA). All organic solvents were HPLC grade and other chemicals were of analytical grade. Deionized (DI) water was used for preparation in this study.

### 2.2. AX-Loaded Emulsion Preparation

The oil phase was prepared by dispersing AX (0.05 wt%) in soybean oil (99.95 wt%) by continuously stirring for 4 h, followed by sonication for 2 min (the power and frequency of the ultrasound were 90 W and 40 KHz, respectively) to obtain a clear solution. The aqueous phase (containing NFC or WPI) was prepared by adding NFC at different concentrations of 0.3, 0.5, and 0.7 wt% or WPI at 0.7 wt% in 10 mM of potassium phosphate buffer (pH 7) and adding 0.01 wt% sodium azide (NaN_3_) as an antimicrobial agent. The aqueous phase was stirred at 600 rpm for 4 h and sonicated for 10 min to ensure the proper dispersion of NFC or complete dissolution of WPI in the buffer. The emulsions were then prepared by mixing 10 wt% of the oil phase and 90 wt% of the aqueous solution using a homogenizer (HG-15A, DAIHAN Scientific Co., Ltd., Wonju-si, Republic of Korea) at 7500 rpm, while continuously adding the oil phase, followed by 12,000 rpm for 2 min to prepare coarse emulsions. The coarse emulsions were then passed through an ultrasonicator (Biosafer650-92, Nanjing Gengchen Scientific Instrument Co., Ltd., Nanjing, China) at a pressure amplitude of 95% for 10 min in a pulse mode (3 s on and 2 s off). The resulting fine emulsions were then stored in a 50 mL test tube at 4 °C before further analysis. The emulsions were prepared in triplicate.

### 2.3. Gastrointestinal Tract Model (GIT Model)

The gastrointestinal fate of the emulsion delivery systems was determined using a simulated gastrointestinal tract model based on an INFOGEST static in vitro simulation of gastrointestinal food digestion [31]. The mean particle size, ζ-potential values, microstructure, and AX content of all emulsions after each stage in the GIT model were determined according to the methods described above.

For the initial stage, all samples were diluted with 10 mM of potassium phosphate buffer (pH 7) to achieve an appropriate concentration of 2 wt% oil. The samples were then placed in a 100 mL glass beaker and incubated in an incubator shaker (SWB-20L, Stirred Water Bath, Major Science Co., Ltd., Saratoga, CA, USA) at 37 °C for 2 min to mimic the human GIT model.

For the oral stage, 20 mL of simulated saliva containing 0.03 g/mL of mucin was preheated to 37 °C for 5 min and then mixed with the initial samples. The pH of the mixture was adjusted to 6.8 with HCl and NaOH and then the mixture was incubated at 37 °C with continuous agitation at 100 rpm for 5 min to mimic oral conditions. Then, the mixture was collected, and 20 mL of the mixture (bolus) was further used for the gastric stage and the rest was used for analysis.

For the gastric stage, simulated gastric fluid (SGF) was prepared by dissolving 2 g of sodium chloride (NaCl) and 7 mL of hydrochloric acid (HCl) in double-distilled water. The total volume of SGF was adjusted to 1 L. Twenty mL of “bolus” samples from the oral phase were mixed with 20 mL of SGF containing 0.0032 g/mL of pepsin. The pH of the mixture was adjusted to 2.5 with HCl and NaOH, and the samples were then incubated at 37 °C with continuous agitation at 100 rpm for 2 h to mimic the gastric stage. At the end of the gastric stage, the mixture was collected, and 30 mL of the mixture (chyme) was further used for the intestinal stage and the rest was used for analysis.

For the small intestine stage, 30 mL of “chyme” samples from the gastric stage were placed in a glass beaker in a water bath at 37 °C. The samples were continuously swirled at 100 rpm and their pH was adjusted to 7.0. Then, 1.5 mL of simulated intestinal fluid (SIF) containing 0.25 M CaCl_2_ and 3.75 M NaCl, and 3.5 mL of bile salt solution were prepared at 37 °C and added to the glass beaker. The pH of this system was readjusted to 7.0. A porcine pancreatin solution containing pancreatic lipase (2.5 mL) was prepared and preheated at 37 °C. Then, the porcine pancreatin solution was added to the glass beaker and an automatic pH-stat titration instrument (Metrohm, USA Inc., Riverview, FL, USA) was used to monitor and maintain the pH at 7.00 by titrating 0.1 M NaOH solution into the glass beaker for 2 h to mimic the small intestine stage. The temperature was maintained at 37 °C during this stage. Then, after the 2 h incubation, the samples were collected in the glass beaker for further analysis. The number of moles of NaOH titrated to neutralize the mixture from the small intestine was used to calculate the amount of free fatty acids (FFAs) released from the system using this equation:(1)$$\mathrm{FFA}\;\left(\%\right)=\frac{V_{\mathrm{NaOH}}\;\times \;m_{\mathrm{NaOH}}\;\times \;M_{\mathrm{lipid}}}{W_{\mathrm{lipid}}\;\times \;2}\;\times 100$$
where *V_NaOH_* is the volume of NaOH (L) used to neutralize the FFAs produced in the small intestinal stage, *m_NaOH_* is the molarity of the NaOH solution (0.1 M), *M_lipid_* is the molecular weight of soybean oil (920 g/mol), and *W_lipid_* is the total weight of oil in the small intestinal stage (0.15 g).

All emulsions were kept after passing through each stage to study the mean particle diameter, ζ-potential values, and microstructure after in vitro digestion.

After the samples passed through the small intestine stage, “digesta fraction” was collected for the determination of AX content in the digesta phase. “Micelle fraction” was prepared by centrifuging the digesta fraction at 1968× *g* for 30 min at room temperature. Following centrifugation, a clear supernatant was collected for the determination of AX content in the micelle phase.

### 2.4. Storage Stability

Each emulsion sample (10 g) was placed into a cylindrical emulsion tube (20 × 70 mm). All tubes were stored at room temperature (25 °C) in dark conditions for 30 days. The stability of the emulsions was evaluated at 1 and 30 days. All emulsions were kept for determining the mean particle size, ζ-potential values, viscosity, and encapsulation efficacy.

### 2.5. Microstructure

The microstructures of AX-loaded emulsions and digested samples after passing through each stage of the gastrointestinal tract (GIT) model were determined using a confocal laser scanning microscope according to Jain et al., 2020, with modifications. Samples were mixed with a Nile red solution dispersed in ethanol at 0.01% *w*/*v*. An aliquot (15 µL) of the mixed solutions of the samples and Nile red solution was placed on a microscopic glass slide and gently covered with a coverslip. The samples were observed with an FV1000 confocal laser scanning microscope (Olympus Corporation, Tokyo, Japan) using a 40× objective lens. The excitation wavelength was set to 488 nm to scan the images.

### 2.6. Physicochemical Properties of Emulsions

#### 2.6.1. Determination of AX Content

To determine the chemical stability of AX, which was encapsulated in the emulsions, the AX content in the emulsions was determined using a UV-Vis spectrophotometer (UV-1601 SHIMADZU, Shimadzu Co., Kyoto, Japan). The non-encapsulated AX was removed by adding dichloromethane to the emulsions at a ratio of 1:2, gently mixing it, and discarding the solvent phase (upper part), which contained the non-encapsulated AX. The encapsulated AX in the emulsions was extracted from the emulsions using a mixture of dichloromethane/methanol (1:1, *v*/*v*) added to the emulsions at a ratio of 1:1 (*v*/*v*), and then the mixtures were vigorously vortexed for 30 s. The samples were centrifuged at 1968× *g* for 5 min at 25 °C. Then, dichloromethane layers were collected, and this procedure was repeated 3 times to extract the remaining AX from the emulsions. The collected layers were examined using a UV-Vis spectrophotometer at 474 nm. The content of AX was then calculated using a standard curve (R^2^ = 0.994). Measurements were performed in triplicate.

#### 2.6.2. Encapsulation Efficiency (EE)

The emulsions were extracted as previously described in Section 2.6.1, and EE was measured using the equation below:(2)$$\mathrm{EE}\;\left(\%\right)=\frac{\mathrm{Amount}\;\mathrm{of}\;\mathrm{AX}\;\mathrm{containing}\;\mathrm{in}\;\mathrm{sample}}{\mathrm{Amount}\;\mathrm{of}\;\mathrm{AX}\;\mathrm{added}\;\mathrm{initially}}\;\times 100$$

#### 2.6.3. Particle Size Measurements

The particle size and size distribution of all the samples were measured using a laser diffraction particle size analyzer (PSA 1190; Anton Paar GmbH, Graz, Austria). All samples were diluted with 10 mM of potassium phosphate buffer (pH 7), except for the emulsion obtained after passing through the gastric stage conditions, which was diluted with DI water (pH 2.5) to avoid multiple scattering effects. The refractive indices of the oil and aqueous phases were set to 1.46 and 1.33, respectively, for the calculation. Particle size was expressed as the surface-weighted mean diameter (d_32_).

#### 2.6.4. ζ-Potential Measurements

The ζ-potential value of the AX-loaded emulsions was measured using a particle electrophoresis instrument (Zetasizer Nano ZS, Malvern Instruments Ltd., Malvern, Worcestershire, UK). All samples were diluted with 10 mM of potassium phosphate buffer (pH 7), except for the emulsion obtained after passing through the gastric stage conditions, which was diluted with DI water (pH 2.5) to avoid multiple scattering effects.

#### 2.6.5. Color Measurement

The color of the AX-loaded emulsions was measured in terms of *L** (lightness), *a** (red to green), and *b** (yellow to blue) using a colorimeter (Color Reader CR -20; Konica Minolta Sensing Americas, Inc., Ramsey, NJ, USA). The colorimeter was first calibrated with a standard white plate before use. Then, the recorded values of *L**, *a**, and *b** were used to investigate the total color change (Δ*E*) using the following equation:(3)$$∆E=\sqrt{{\left(L^{*}-L_{i}^{*}\right)}^{2}+{\left(a^{*}-a_{i}^{*}\right)}^{2}+{\left(b^{*}-b_{i}^{*}\right)}^{2}}$$
where *L*, a*,* and *b** are the color parameters measured at time t and *L_i_*, a_i_*,* and *b_i_** are the color coordinates measured after the preparation of the emulsions.

#### 2.6.6. Viscosity Measurements

The viscosity of the emulsions was measured using a rheometer (HAAKE™ MARS™ 40 Rheometers, Thermo Fisher Scientific Inc., Agawam, MA, USA) equipped with a cone and plate sensor (2° cone angle, made of Titanium, and 0.05 mm gap). The steady-flow tests were conducted to determine the viscosity of the emulsions. The samples were continuously sheared from 0.1 to 300 s^−1^ in 3 min, followed by a decrease from 300 to 0.1 s^−1^ in 3 min. The temperature was set at 25 °C through this experiment.

### 2.7. Determination of AX Content after the GIT Model

After the samples passed through the small intestine stage, 10 mL of the samples were collected after the small intestine stage and placed in a 15 mL centrifuge tube. The collected samples were centrifuged at 1,968 × *g* for 30 min at room temperature. Following centrifugation, a micelle fraction, a clear supernatant, was collected, in which AX was dissolved. A clear supernatant (2 mL) of the micelle fraction was mixed with a mixture of dichloromethane/methanol (1:1, *v*/*v*) added to the solution at a ratio of 1:1 (*v*/*v*), and then the mixtures were vigorously vortexed for 30 s. The samples were centrifuged at 1968× *g* for 5 min at 25 °C. Then, dichloromethane layers were collected, and this procedure was repeated 3 times to extract the remaining AX from the solution. The collected layers were examined using a UV-Vis spectrophotometer at 474 nm. The bioaccessibility and stability of AX were calculated using the following equations:(4)$$\mathrm{Bioaccessibility}\;\left(\%\right)=\left(C_{\mathrm{Micelle}}/C_{\mathrm{Digesta}}\right)\times 100$$
(5)$$\mathrm{Stability}\;(\%)=\left(C_{\mathrm{Digesta}}/C_{\mathrm{Initial}}\right)\times 100$$
where C_Initial_, C_Micelle_, and C_Digesta_ are AX concentrations primarily added to the emulsions, in the micelle fraction phase, and in the digesta fraction phase following in vitro digestion, respectively.

### 2.8. Statistical Analysis

All the data were collected in triplicate and are reported as the mean ± standard deviation. Analysis of variance (ANOVA) followed by Tukey’s post hoc test were used to determine the significant differences among the mean values. A significant difference was accepted at *p* ≤ 0.05. A paired-sample t-test was used to determine the significant differences between the freshly prepared emulsions and those obtained after stored for 30 days. A significant difference was accepted at *p* ≤ 0.05. A statistical analysis was calculated using statistical software (SPSS version 26.0, SPSS Inc., Chicago, IL, USA).

## 3. Results and Discussion

### 3.1. Emulsion Properties after Storage for 30 Days

The surface-weighted mean droplet diameters (d_32_) of the freshly prepared and 30-days-stored AX-loaded emulsions stabilized by different NFC concentrations were determined and are shown in Table 1. A slight increase in the mean particle diameters of NFC- stabilized emulsions was observed when the NFC concentration was increased from 0.3 to 0.7 wt%. During emulsification, the emulsions stabilized by NFC had a different mechanism than the solid particles with a low aspect ratio [32]. The mechanism of NFC formation at the O/W interface was slow adsorption on the droplet surfaces to form thick interfacial layers between the emulsion droplets and continuous phase. This was attributed to the physical properties of NFC, which is a long fiber that can be absorbed on the droplet interfaces during homogenization and build up tangibly crowded interfacial layers with NFC [28]. However, no significant difference (*p* > 0.05) was observed in the particle diameters when the concentration of NFC was increased. This indicates that the high concentration of NFC has no effect on the mean particle size of the emulsion droplets of AX-loaded emulsions. Interestingly, no significant differences (*p* > 0.05) were observed between the d_32_ values of the initial emulsions and emulsions after 30 days of storage. This result is similar to a previous study [33], in which a Pickering emulsion stabilized with a nanoparticle-based emulsifier was able to protect the oil droplets from aggregation due to a fibrous structure formation in the aqueous phase. Therefore, the emulsions stabilized with NFC in this study also have the potential to improve the long-term stability of emulsions.

The ζ-potential values of all emulsions showed a strong negative charge originating from the negative carboxyl groups (-COO^−^) at the interfaces. The values obtained were lower than −20 mV. The higher the absolute values of the ζ-potential, the greater the electrostatic repulsion between the droplets, resulting in better stability of the emulsions. It was observed that emulsions with ζ-potential values close to ± 30 mV had sufficient repulsive forces to achieve better physical colloidal stability [34]. Our results suggest that NFC as an emulsifier can prevent the aggregation of droplets due to electrostatic repulsion [35,36]. It was also observed that the ζ-potential values increased when the NFC concentration was increased from 0.3 to 0.7 wt%. This could be due to the closer packing of the NFC at the interfaces of the emulsion droplets, resulting in stronger electrostatic repulsion between them, which could prevent them from coalescing or aggregating. Consequently, the strong electrostatic repulsion at the droplet interface may also allow the emulsions to stabilize over longer periods of time [37,38]. However, a significant (*p* ≤ 0.05) increase in ζ-potential values was observed in all emulsions after 30 days of storage, which may be due to the degradation of AX [14]. This indicates that the composition of the interfaces changes after long-term storage, especially for emulsions with a higher concentration of NFC.

The viscosity of the freshly prepared emulsions increased significantly (*p* ≤ 0.05) with increasing the NFC concentration from 0.3 to 0.7 wt% (Table 1), indicating that the network-like 3D structure of NFC in the aqueous phase formed a large amount of enmeshment of cellulose [39] with the increasing NFC concentration, which prevented the coalescence of droplets. In addition, the increase in NFC concentration could play an important role in improving the stability of the emulsions due to an increase in the yield stress (Figure 1) of an aqueous phase in the emulsions leading to a stronger 3D network in the aqueous phase [32] and other benefits resulting from the increase in NFC concentration for the encapsulation of the lipid droplets [40,41,42]. There were no significant differences (*p* > 0.05) in the viscosity of the original emulsions and the emulsions stored for 30 days, except for the emulsions stabilized with 0.5 wt% NFC, suggesting that the physical properties of AX-loaded emulsions depend on the emulsifier concentration [43]. Therefore, the NFC-stabilized emulsions without AX were prepared as the control (see Appendix A).

The encapsulation efficiency (EE) of AX-loaded emulsions was determined and is shown in Table 1. A significant increase (*p* ≤ 0.05) in the EE values of the freshly prepared emulsions and emulsions stored for 30 days was observed when the NFC concentration increased from 0.3 to 0.7 wt%. This result is in agreement with other studies [44,45] in which nanocellulose was used as an emulsifier, which tends to form a 3D structure in the aqueous phase, which may increase the stability of the emulsion and prevent the loss of AX with an increasing NFC concentration. Therefore, our studies suggest that the formation of a 3D network of added NFC acts as a physical barrier that can prevent degradation. Therefore, it was observed that the EE value of the NFC-stabilized emulsion decreased significantly (*p* ≤ 0.05) after 30 days of storage with a decreasing NFC concentration, which was due to thin interfacial layers of emulsifier formed, leading to a weak physical barrier in aqueous phase [16,46]. In addition, there were further changes that showed AX could react with its surroundings, which favored the degradation of AX following storage. Moreover, previous studies showed that the EE value of AX was only 58.76% of emulsions stabilized by sodium carboxymethyl cellulose and microcrystalline cellulose [1]. Therefore, NFC could be a potential emulsifier for AX-loaded emulsions to partially protect them from chemical degradation.

The color of the freshly prepared NFC-stabilized emulsions with different NFC concentrations from 0.3 to 0.7 wt% and of the emulsions after 30 days of storage were determined and shown in Table 2. The lightness (*L**) does not significant change when increasing the NFC concentration. In addition, this result is in good agreement with d_32_, which cannot observe the change in d_32_ for the system when increasing the NFC concentration (Table 1.). An increasing of droplet size tends to decrease *L** of the emulsions due to light being absorbed to a great extent when the light-scattering efficiency decreases. Therefore, emulsions with large oil droplets tended to have lower *L** values than small oil droplets [32,47,48]. Moreover, *L** decreased significantly (*p* ≤ 0.05) after the emulsions were stored for 30 days. This was due to the flocculation of oil droplets, which resulted in emulsion droplet aggregates that occurred in the emulsion-based delivery system using NFC as an emulsifier [29]. Therefore, the emulsion system stabilized by NFC generally produced large emulsion droplets due to the long and fibrous properties of NFC, which may have induced the flocculation rather than coalescence of the emulsion droplets [49].

The redness (*a**) and yellowness (*b**) of all emulsions decreased significantly (*p* ≤ 0.05) after 30 days of storage due to the degradation of AX [37,50]. The determination of color values in terms of total color changes (∆E) is more convenient to compare the color change between emulsions. The total color changes decreased with the increasing NFC concentration, indicating that the lower total color change values indicate the better ability of the NFC-stabilized emulsion to protect AX from chemical degradation, which is in good agreement with the encapsulation efficiency results of AX-loaded emulsions (Table 1). The total color changes of the emulsions after 30 days of storage are also less than 3.0, which is in good agreement with the appearance of the emulsions (Figure 2); therefore, the difference could not be detected by the human eye.

### 3.2. Influence of Emulsifier Concentrations on the Physicochemical Properties of Lipid Droplets in Gastrointestinal Fate

#### 3.2.1. Initial Stage

The surface-weighted mean diameter (d_32_), ζ-potential values, and microstructure were determined to establish the stability of the initial emulsions. The emulsions prepared with WPI as an emulsifier formed smaller emulsion droplets than the emulsions prepared with NFC as an emulsifier (Figure 3a). Therefore, WPI was a potential emulsifier compared with NFC to produce small emulsion droplets during the same homogenization process. On the other hand, NFC was a less effective emulsifier than WPI due to the slowly irreversible adsorption of the particles at interfaces, where their high-adsorption energy can produce thick interfacial layers, which potentially prevents droplet aggregation [20,27], leading to the formation of more stable emulsions.

The d_32_ values increased significantly (*p* ≤ 0.05) from 8 to 14 µm when the NFC concentration was increased from 0.3 to 0.7 wt% (Figure 3a). Moreover, the flocculation and slight coalescence of the droplets were promoted by the high NFC concentration, which was confirmed by confocal microscopy images (Figure 4). This result is consistent with previous studies reporting that high NFC concentrations can lead to the formation of flocculation and coalescence of droplets during homogenization, which decreases homogenization efficiency [28]. The ζ-potential value of the emulsions showed a high negative charge, and the magnitude of the ζ-potential was greater than −30 mV for all emulsions (Figure 3b), indicating a greater electrostatic repulsion between the droplets, suggesting that the prepared emulsions had good stability.

#### 3.2.2. Oral Stage

The mean particle diameter (d_32_) of the WPI-stabilized emulsions increased from about 0.8 to 1.7 µm after passing through the oral phase (Figure 3a). The WPI-stabilized emulsions were observed as small fragments of oil droplets with larger dimensions than those of the freshly prepared WPI-stabilized emulsion, as shown by the confocal microscopy images (Figure 4). This suggests that the protein-coated oil droplets may form clusters by bridging flocculation due to their electrostatic interactions with mucin, a negatively charged biopolymer, and depletion flocculation triggered by other noncolloidal salivary particles [43,51]. Moreover, the flocculation of protein-coated droplets after mixing with simulated saliva fluids was confirmed by confocal microscopy images.

Interestingly, there was a slight increase in the mean particle diameter (d_32_) of the NFC-stabilized emulsions after passing through simulated oral conditions (Figure 3a). However, the NFC-coated droplets were able to cluster together by NFC, as shown by the confocal microscopy images (Figure 4).

In addition, a significant decrease in the negative charge was observed in all tested emulsions after passage through the oral conditions (Figure 3b). This decrease could be due to electrostatic screening effects caused by the interaction of mucin with the surface of the oil droplets or by the mineral ions present in the simulated saliva fluid [20,52].

#### 3.2.3. Gastric Stage

After passing through the gastric stage, a significant increase in the mean particle diameter of the WPI-stabilized emulsion was observed (Figure 3a). These results are in good agreement with the confocal microscopy images, which showed large oil-droplet clusters (Figure 4). The increase in the mean particle diameter of the WPI-stabilized emulsions may be due to several reasons. Firstly, the adsorbed proteins may be hydrolyzed by pepsin, resulting in weaker electrostatic repulsion. Secondly, it may be due to depletion or bridging flocculation induced by mucin from the simulated saliva fluid. Thirdly, changes in the pH and ionic strength in the gastric environment may have reduced the electrostatic repulsion between the emulsion droplets [20,29,52]. In contrast, large clusters were observed in the NFC-stabilized emulsion droplets, as indicated by the results of the mean particle diameter (Figure 3a) and confocal microscopy images (Figure 4), which may be attributed to the formulation of NFC in the aqueous phase after the gastric conditions.

The magnitude of the ζ-potential of the protein-absorbed interfaces was relatively low and negative (–4.1 mV) under gastric conditions (Figure 3b). Therefore, the droplets with a low negative charge may be due to mineral counter-ions that could reduce the magnitude of the ζ-potential on the droplets. In addition, the relatively low pH can be attributed to the fact that the WPI-stabilized emulsion droplets were coated with anionic mucin molecules and some of the proteins may have been digested and displaced. Therefore, this effect may neutralize their charge in the gastric stage [53]. In NFC-stabilized emulsions, it was observed that the magnitude of the ζ-potential of the NFC-absorbed interfaces decreased after passing through the simulated gastric fluids, which may be attributed to the changing electrical characteristics of NFC under highly acidic conditions. These results suggest that NFC lose their negative charge at low pH values in the stomach due to the protonation of the carboxyl groups [35].

#### 3.2.4. Small Intestinal Stage

After the samples passed through the small intestine stage, all tested emulsions showed significant changes in their mean particle diameters (Figure 3a), ζ-potential values (Figure 3b), and microstructures (Figure 4). The mean particle diameter increased considerably, which was confirmed both by light scattering (Figure 3a) and confocal microscopy (Figure 4), for which there could be several reasons. Firstly, lipolysis might have led to an accumulation of fatty acids at the emulsion interface, resulting in the displacement of fatty acids, which generally contributes to the coalescence of the droplets [51]. Secondly, a complex mixture of many different types of particles, including undigested oil droplets, undigested protein aggregates, micelles, vesicles, and insoluble calcium salts, formed in the small intestinal phase, which may have contributed to the coalescence of the droplets [43,53]. When the NFC concentration was increased, larger particles remained in the samples, which was due to the insoluble materials in these systems. The remaining particles were visible in the confocal microscopy images following the small intestine stage (Figure 4).

All emulsions exhibited a strong negative charge at the end of the gastrointestinal fate (Figure 3b). The anionic species that were present in the small intestinal phase may have originated from the original emulsions or from the stimulated gastrointestinal fate. For example, micelles, vesicles, calcium salts, fatty acids, and undigested lipids may be associated with the high negative charge of the emulsions [29,53]. However, it was observed that the ζ-potential magnitude for only 0.5 and 0.7 wt% NFC-stabilized emulsions did not show any significant differences (*p* > 0.05) from the initial emulsions (Figure 3b), indicating that an NFC concentration higher than 0.5 wt% may improve the stability of AX-loaded emulsions.

### 3.3. Influence of Emulsifier Concentrations on Fat Digestibility

The influence of emulsifier type and concentration on lipid digestibility was determined using an automatic titration method for 120 min during the small intestine stage (Figure 5). These lipid digestion profiles were used to calculate the initial lipid digestion rate (Figure 6a) and the final extent of lipid digestion (Figure 6b). Specifically, the initial rate of FFA release was assessed by creating a linear line to fit the first five minutes of lipid digestion, while the final extent of digestion was assessed by averaging the last five minutes of lipid digestion.

Initially, a significant increase (*p* ≤ 0.05) in FFA release was observed during the first 10 min of digestion, followed by the gradual digestion of lipid droplets that became relatively constant. It was observed that the WPI-stabilized emulsions exhibited the highest initial rate and final extent of lipid digestibility (Figure 6). This behavior can be attributed to the fact that the small lipid droplets that were digested by lipase had a large surface area. Therefore, the WPI-stabilized emulsions exhibited the highest digestion rate and extent among all the emulsions tested.

In contrast, the initial rate and final extent of NFC-stabilized emulsions were lower than those of the WPI-stabilized emulsions (Figure 6). In addition, it was observed that increasing the concentration of NFC slowed the initial rate and final extent of lipid digestion, with the emulsion stabilized with 0.7 wt% NFC having the slowest initial rate and final extent of lipid digestion (Figure 6). The decrease in lipid digestibility of emulsions was observed with increasing the NFC concentration. This can be attributed to a number of factors. Firstly, increasing the NFC concentration led to an increase in the size of emulsion droplets, which decreased the contact area between the lipase and oil droplets and reduced lipolysis [54]. Secondly, the increasing concentration of NFC led to an increase in viscosity, as shown in Table 1. When the NFC concentration was increased from 0.3 to 0.7 wt%, it resulted in a four-times-higher emulsion stability compared to the lowest NFC concentration. However, a significant increase in viscosity may have led to the formation of a disordered network of NFC surrounding the lipid droplets, which may hinder the access of lipase and bile salts to the droplet surfaces [55]. Thirdly, NFC could bind to the calcium ion or bile salts, which can reduce the release of FFAs [49]. Therefore, NFC may have partially inhibited lipid digestion. This result agreed well with the microstructure observed through confocal microscopy images of the samples after passing through the small intestine conditions (Figure 4), suggesting that the remaining lipid droplets for NFC-stabilized emulsions were present at the end of the gastrointestinal fate.

NFC as a natural emulsifier can potentially provide resistance to AX from harsh conditions without being fully released at the end of the gastrointestinal fate. This result could be highlighted for future studies to develop encapsulated AX to provide potential health benefits [43].

### 3.4. Influence of Emulsifier Concentrations on AX Bioaccessibility

AX generally has poor bioaccessibility properties due to its low chemical stability and solubility in gastrointestinal fluids [6]. Therefore, the bioaccessibility and stability of AX with different emulsifier concentrations were determined to compare the bioaccessibility and stability of AX. Stability is the percentage of AX in the digesta phase after passing through the simulated GIT fate compared to the initial value, while bioaccessibility is the percentage of AX in the mixed-micelle phase compared to the total digesta phase after 2 h of incubation.

The AX content in the mixed-micelle and total digesta phases was determined after the small intestine stage (Figure 7). The initial amount of AX added to the emulsions was 5 mg. As the NFC concentration increased, the AX bioaccessibility decreased, which may have been due to the NFC formulation that can inhibit lipolysis. Therefore, the oil phase that remained entrapped in the NFC structures may have contained AX. In addition, the insoluble aggregates may also entrap the mixed micelles containing AX, leading to a decrease in bioaccessibility [11].

In addition, the stability of AX was observed to increase with the increasing NFC concentration. The increase in stability could be due to a stronger physical barrier provided by NFC as a physical barrier for the AX molecules. Higher NFC concentrations promoted the protection of AX from degradation under severe conditions in the gastrointestinal model. However, the stability of the WPI-stabilized emulsions was poorer than that of the NFC-stabilized emulsions (Figure 7), suggesting that WPI as an emulsifier may have promoted the chemical degradation of AX because of the increase in the surface area leading to a decrease in the aggregation in the emulsion droplets. However, the stability of all emulsions is relatively low (< 51%), which is due to the fact that the EE of all emulsions from the freshly encapsulated emulsion was low (EE < 70%); therefore, they presented low stability after passing through the harsh environment of the gastrointestinal tract. However, the WPI-stabilized emulsions had high AX bioavailability, suggesting that they were digested more rapidly due to the smaller droplets [19].

## 4. Conclusions

In this study, NFC-stabilized emulsions with different concentrations were successfully evaluated for their stability over one month and simulated in vitro gastrointestinal behavior. NFC was observed to be potentially used as a plant-based emulsifier to stabilize emulsions containing AX. However, NFC produced large emulsion particles when compared to the WPI-stabilized emulsion that possibly induced the lipid droplets held together as large clusters. Therefore, this phenomenon can improve the emulsion stability by inhibiting coalescence and forming an NFC network in the aqueous phase, resulting in higher viscosity at high NFC concentrations.

Fat digestibility and AX bioaccessibility in the emulsions were also determined and increasing the NFC concentration resulted in a decrease in both fat digestibility and bioaccessibility. This suggests that NFC may play an important role in inhibiting fat digestibility by altering the accessible surface area for lipases to digest lipid droplets. As a result, NFC at a low concentration (0.3 wt% NFC) would be useful to release bioactive compounds for absorption due to high bioaccessibility, as it produced small emulsion droplets that had large interfacial surfaces for lipid droplet digestion. In contrast, the use of NFC at a high concentration (0.7 wt% NFC) would be an undesirable concentration to deliver bioactive compounds to the small intestine for absorption. In addition, the use of NFC at high concentrations would be useful for inhibiting or altering lipid digestion for controlling the fat release of foods to control satiety.

For future studies, this study may provide increased health information to investigate other delivery systems using NFC as an emulsifier to improve the stability of astaxanthin and to conduct in vivo studies in the future.

## Figures and Tables

**Figure 1 polymers-15-00901-f001:**
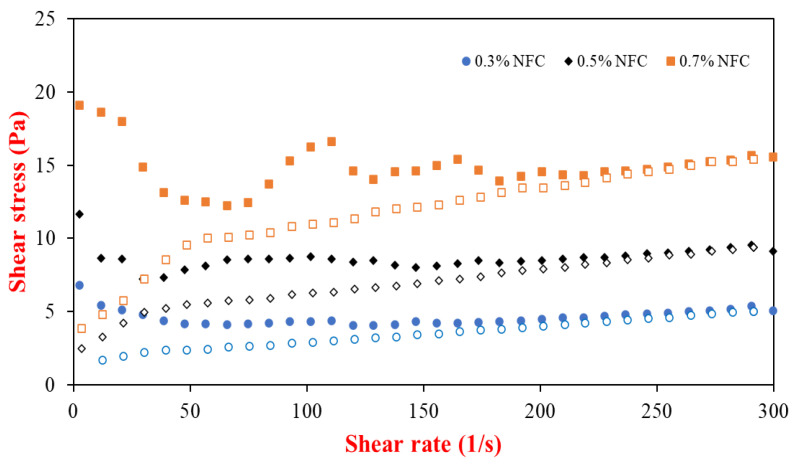
Flow curves of emulsion stabilized by NFC at various concentrations, in which closed symbols represent the upward curves and open symbols represent downward curves.

**Figure 2 polymers-15-00901-f002:**
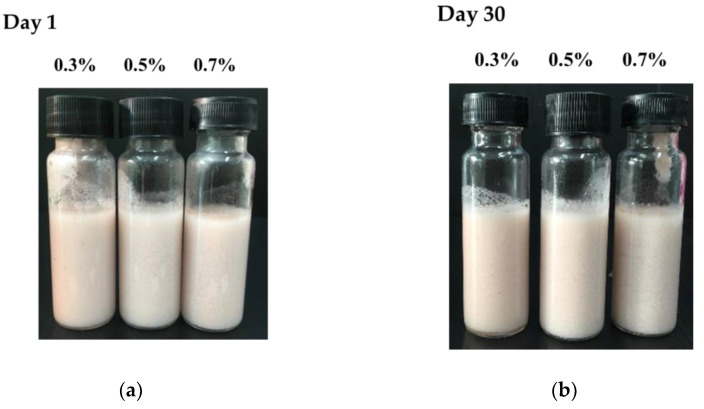
The appearance of the tested emulsions: (**a**) appearance of the freshly prepared AX-loaded emulsions; (**b**) AX-loaded emulsions after 30 days of storage.

**Figure 3 polymers-15-00901-f003:**
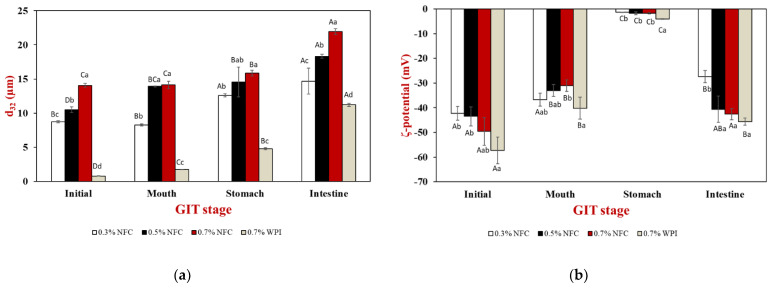
Effects of emulsifier concentrations on (**a**) mean particle diameter (d_32_) and (**b**) ζ-potential values of AX-loaded emulsions. Different uppercase letters (A–C) indicate significant (*p* ≤ 0.05) comparisons between different GIT stages (same emulsifier concentrations). Different lowercase letters (a–d) indicate significant (*p* ≤ 0.05) comparisons between different emulsifier concentrations (same GIT stage).

**Figure 4 polymers-15-00901-f004:**
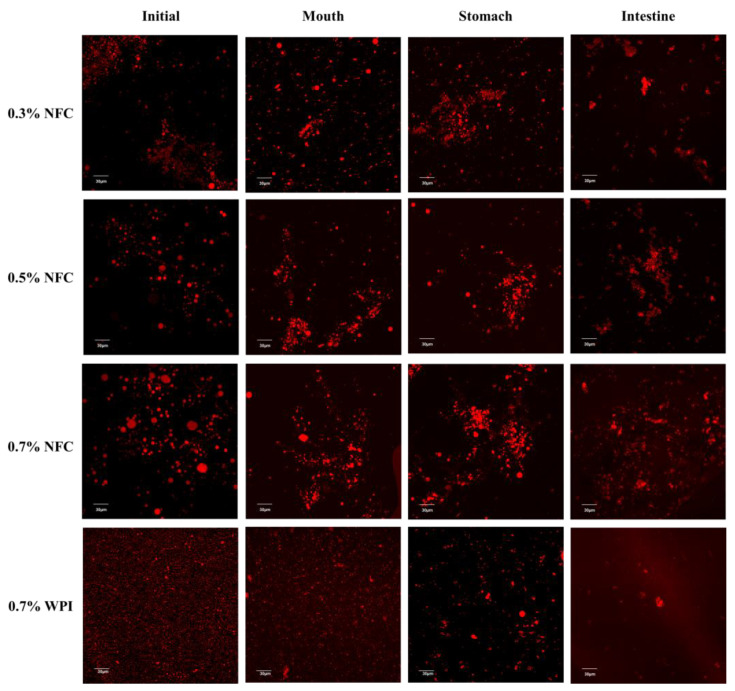
Confocal microscopy images of AX-loaded emulsions prepared with different emulsifier concentrations. The scale bar is 30 µm.

**Figure 5 polymers-15-00901-f005:**
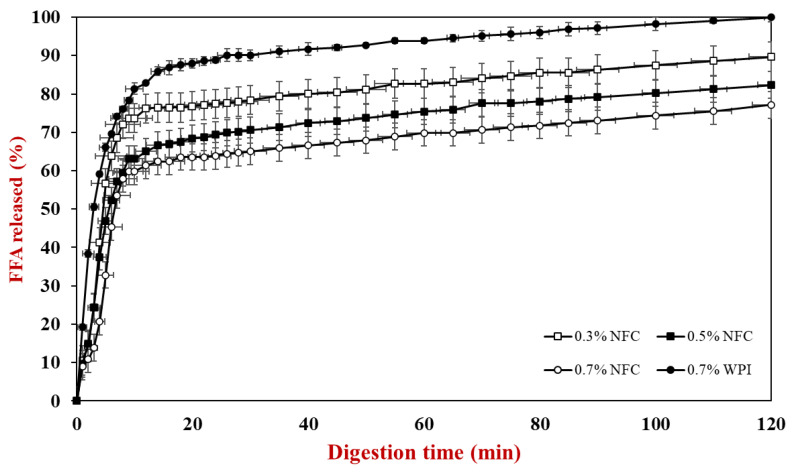
Release of free fatty acids (FFAs) from AX-loaded emulsions under small intestine conditions prepared with various emulsifiers containing nanofibrillated cellulose and whey protein.

**Figure 6 polymers-15-00901-f006:**
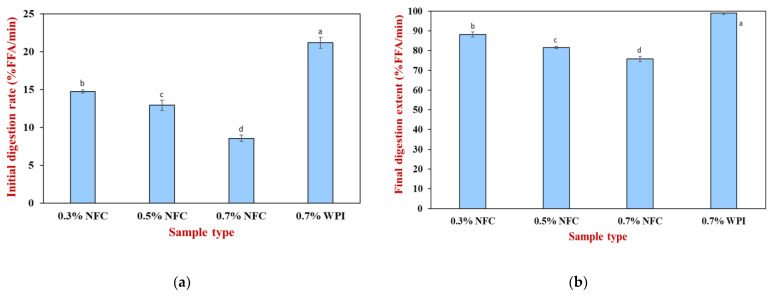
(**a**) Initial digestion rate; (**b**) final digestion extent of AX-loaded emulsions prepared with nanofibrillated cellulose and whey protein. Different lowercase letters (a–d) determine a significant difference in the samples (Tukey, *p* ≤ 0.05).

**Figure 7 polymers-15-00901-f007:**
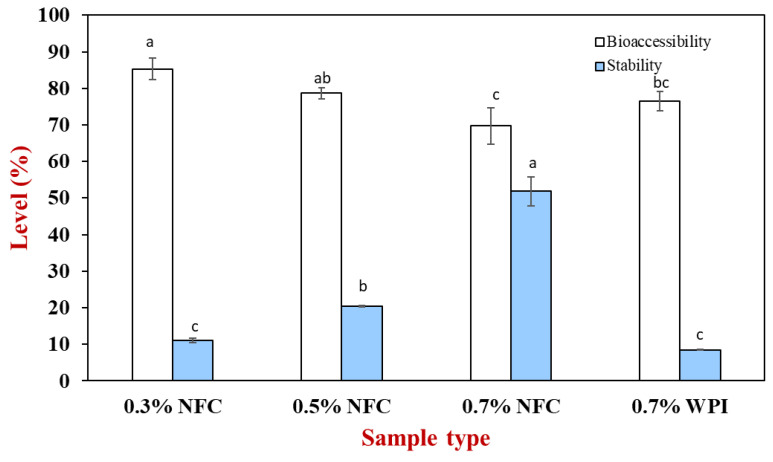
Stability and bioaccessibility in AX-loaded emulsions prepared with NFC and whey protein after undergoing simulated GIT fate. Different lowercase letters (a–c) determine a significant difference in the samples (Tukey, *p* ≤ 0.05).

**Table 1 polymers-15-00901-t001:** Emulsion properties and stability of AX-loaded emulsions that were measured for freshly prepared emulsions and those after being stored at room temperature for one month.

NFC Concentration (%)	d_32_ (µm)		ζ–Potential (mV)		*η*_a,100_ (mPa.s)		Encapsulation Efficacy (%)	
	Day 1	Day 30	Day 1	Day 30	Day 1	Day 30	Day 1	Day 30
0.3	12.65 ± 0.69 a,A	13.59 ± 1.06 a,A	−21.88 ± 3.35 b,A	–25.92 ± 2.85 a,B	41 ± 0.003 a,C	42 ± 0.001 a,C	56.33 ± 0.08 a,B	24.61 ± 0.01 b,C
0.5	13.44 ± 0.01 a,A	14.31 ± 0.70 a,A	−20.58 ± 4.04 b,A	–28.48 ± 3.24 a,B	84 ± 0.004 b,B	96 ± 0.008 a,B	53.75 ± 0.08 a,C	42.60 ± 0.05 b,B
0.7	14.40 ± 1.37 a,A	14.41 ± 1.48 a,A	–25.58 ± 4.35 b,A	–40.33 ± 3.97 a,A	158 ± 0.006 a,A	132 ± 0.017 a,A	70.94 ± 0.01 a,A	47.71 ± 0.05 b,A

Values are measured in triplicate and shown as mean ± standard deviation. Different lowercase letters (a–b) in the same row result in a significant (*p* ≤ 0.05) difference with respect to storage time. Different uppercase letters (A–C) in the same column result in a significant (*p* ≤ 0.05) difference with respect to NFC concentration.

**Table 2 polymers-15-00901-t002:** Color (*L**, *a**, *b**) of AX-loaded emulsions measured after 1 and 30 days of storage at room temperature.

NFC Concentration (%)	Color							
	*L** (Lightness)		*a** (Redness)		*b** (Yellowness)		Δ*E* (Total Color Change)	
	Day 1	Day 30	Day 1	Day 30	Day 1	Day 30	Day 1	Day 30
0.3	54.20 ± 0.17 a,B	52.63 ± 0.12 b,C	9.03 ± 0.12 a,B	7.47 ± 0.06 b,AB	6.17 ± 0.15 a,A	5.43 ± 0.06 b,A	–	2.35 ± 0.13 b,A
0.5	56.77 ± 0.06 a,A	56.20 ± 0.01 b,A	9.10 ± 0.01 a,AB	7.50 ± 0.01 b,A	6.00 ± 0.01 a,A	5.40 ± 0.01 b,A	–	1.80 ± 0.02 b,B
0.7	54.40 ± 0.10 a,B	53.80 ± 0.01 b,B	9.23 ± 0.06 a,A	7.37 ± 0.06 b,B	5.57 ± 0.06 a,B	4.43 ± 0.06 b,B	–	2.27 ± 0.12 b,A

Values are measured in triplicate and shown as mean ± standard deviation. Different lowercase letters (a–b) in the same row result in a significant (*p* ≤ 0.05) difference with respect to storage time. Different uppercase letters (A–C) in the same column result in a significant (*p* ≤ 0.05) difference with respect to NFC concentration.

## Data Availability

Not applicable.

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
