# Peer review of "Astaxanthin-Loaded Pickering Emulsions Stabilized by Nanofibrillated Cellulose: Impact on Emulsion Characteristics, Digestion Behavior, and Bioaccessibility"

_polymers, 2023, doi:10.3390/polym15040901_

Round 1

Reviewer 1 Report

In this paper, authors systematically investigated the effects of emulsifier types and concentrations on NFC-stabilized emulsions. NFC could potentially be used as a plant-based emulsifier to stabilize emulsions containing AX. This investigation can be helpful to obtain an in-depth knowledge of the Fat digestibility and AX bioaccessibility in the emulsions. However, it’s worth noting that some points need to be discussed in the text. Hence, some specific comments are given in the following:

1. In the section of ‘2.5.2. Encapsulation Efficiency (EE)’, how to separate encapsulated AX and unencapsulated AX in the emulsions before analyzing the content of AX? Authors should describe in detail the process in this section.

2. In the section of ‘3. Results and discussion’, where is Figure 3?

3. In the section of ‘Oral stage’ and Figure 2a, d32 slightly increased only for 0.5% NF, why?  

Author Response

We thank the reviewer for the constructive comments and suggestions on our manuscript. An itemized list of changes (in red) that have been made to the manuscript in response to the reviewer’s comments are given in the attached file

Reviewer 2 Report

The article by Sae-chio et al. examines the use of nanocellulose to produce direct Pickering emulsions containing astaxanthin in the oil phase. The authors determine the size of emulsion droplets, their zeta potential, effective viscosity of emulsions, and their color characteristics. The authors do this both for fresh emulsions and after they have been stored or placed in specific conditions simulating the digestive tract. Further, the authors determine the encapsulation efficacy of emulsions with respect to astaxanthin, release of free fatty acids, digestion rate, stability, and bioaccessibility of emulsions. In general, it is a very extensive, interesting, and highly original work. Nevertheless, the article must be revised before being accepted for publication.

Specific comments are as follows.

Line 71: “and a long fibrous structure that potentially forms a 3D network after dispersing in the water phase of emulsions”. This statement needs to be supported by a reference to the literature, e.g., 10.1021/acs.energyfuels.0c02797.

Line 73: “the emulsions stabilized with NFC tend to have high shear viscosity,” These emulsions have a yield stress, and it is responsible for the high sedimentation stability of Pickering nanocellulose-stabilized emulsions (see the reference above).

Line 93: “Nanofibrillated cellulose (NFC) was purchased…” The authors should provide the characteristics of the nanocellulose. Was it a powder or an aqueous dispersion? What was the mass fraction if it was a dispersion? What was the diameter and length of the fibrils of this nanocellulose? Perhaps the authors could have provided SEM images of the nanocellulose particles.

Line 100: “followed by sonication for 2 min”. How exactly was the system sonicated? Was it an immersion waveguide or an ultrasonic bath? What power and frequency of ultrasound was used?

Line 104: “to ensure proper dissolution of NFC in the buffer”. Nanocellulose does not dissolve under these conditions. It disperses.

Lines 203-208: “equipped with a cone and plate sensor”, “and gap between the upper plate and lower plate of measurement”. The authors contradict themselves when they write about the cone-plate system in the first case and the plate-plate system in the second case.

Line 222: “1968×g”. There is probably a comma missing here.

Line 240: “A slight increase in droplet diameters was observed when the NFC concentration was increased from 0.3 to 0.7 wt%.” The authors should specify the effective diameter of NFC particles in the droplet-free NFC dispersion. In other words: Why do the authors think that they measure the size of the droplets rather than the size of the cellulose particles (or their aggregates)?

Line 266: A replacement is needed: “greater than -20 mV. The higher the values of ζ-potential” -> “lower than -20 mV. The higher the absolute values of ζ-potential”.

Line 280: “The viscosity of the freshly prepared emulsions increased significantly”. Nanocellulose-stabilized emulsions have a strong non-Newtonian behavior, and their viscosity is strongly dependent on shear rate. Neither Table 1 nor the text states under which conditions the viscosity of the emulsions was measured for comparison. Due to the strong dependence of viscosity on shear rate, a more unambiguous characteristic of nanocellulose dispersions or nanocellulose-stabilized emulsions is their yield stress. Moreover, nanocellulose dispersions can exhibit wall slip instead of flow at high shear rates. It would be more informative if the authors gave flow curves of emulsions rather than viscosities measured at a single shear rate.

Line 281: “indicating that the network-like 3D structure of NFC in the aqueous phase increased with increasing NFC concentration”. There is no such thing as "an increase” in “network-like 3D structure of NFC”. In the dispersion of nanocellulose, its fibrils form entanglements, and an increase in the concentration of nanocellulose leads to an increase in the number of entanglements and consequently an increase in the effective viscosity (see, e.g., 10.1016/j.triboint.2022.108080).

Line 283: “the change in NFC concentration could play an important role in improving the stability of the emulsions due to the increased viscosity of the aqueous phase”. It is more correct to indicate that there is an increase in the yield stress of the aqueous medium rather than in its viscosity. In contrast to the yield stress, an increase in the viscosity cannot increase stability: it can only slow down the breakdown of the emulsion.

Line 297: “our studies suggest that the formation of a 3D network of added NFC”. The authors do not show in their work the formation of a 3D network from NFC. They do not use flow curves or viscoelasticity studies. The authors should use weaker expressions.

Line 313: “it is known that large oil droplets have lower light scattering compared to small oil droplets”. This is not true. According to the Rayleigh scattering theory, large droplets scatter light many times more strongly than small droplets. In the case of large micron-sized droplets, there is reflection and refraction of light rather than its scattering.

Line 318: A replacement is needed: “This was due to the flocculation of oil droplets, which resulted in larger emulsion droplets” -> “This was due to the coalescence of oil droplets, which resulted in larger emulsion droplets” or “This was due to the flocculation of oil droplets, which resulted in larger emulsion droplet aggregates”. Flocculation of droplets does not lead to an increase in their size (the size of their aggregates increases).

Line 373 and further: “Figure 3”. The article does not contain figure 3.

Author Response

We thank the reviewer for the constructive comments and suggestions on our manuscript. An itemized list of changes (in red) that have been made to the manuscript in response to the reviewer’s comments are given in the attached file.

Round 2

Reviewer 1 Report

No any comments.

Author Response

Dear Reviewer,

We thank the reviewer for accepting our manuscript and all constructive comments. 

Best Regards,

Thunnalin Winuprasith

Reviewer 2 Report

The authors have made all the necessary corrections to the manuscript, and it can now be published. The only point I would like to make is that the resolution of the figures in the article has deteriorated. In the first version of the manuscript, the figures were sharper.

Author Response

Dear Reviewer,

We thank the reviewer for the constructive comments and suggestions on our manuscript. The resolution of all figures was increased.

Best Regards,

Thunnalin Winuprasith